# Relationship between Starting Torque and Thermal Behaviour for a Permanent Magnet Synchronous Generator (PMSG) Applied with Vertical Axis Wind Turbine (VAWT)

**Mintra Trongtorkarn [1], Thanansak Theppaya [2], Kuaanan Techato [3], Montri Luengchavanon [4],\* and Chainuson Kasagepongsarn [5]**

[1] Energy Technology Program, Faculty of Engineering, Prince of Songkla University, Hatyai, Songkhla 90110, Thailand; mintra.nan19@gmail.com
[2] Department of Mechanical Engineering, Faculty of Engineering, Prince of Songkla University, Hatyai, Songkhla 90110, Thailand; thanansak.t@psu.ac.th
[3] Environmental Assessment and Technology for Hazardous Waste Management Research Centre, Faculty of Environmental Management, Prince of Songkla University, Hat Yai, Songkhla 90110, Thailand; kuaanan.t@psu.ac.th
[4] Wind Energy and Energy Storage Systems Centre (WEESYC), Faculty of Environmental Management, Prince of Songkla University, Hatyai, Songkhla 90110, Thailand
[5] Renewable Energy and Environmental Research for Local Community Unit (REERCU), Surat Thani Rajabhat University, Mueang, Surat Thani 84100, Thailand; chainuson.kas@sru.ac.th
\* Correspondence: montri.su@psu.ac.th

**Abstract:** The application of wind turbine technology in low wind speed regions such as Southeast Asia has recently attracted increased attention. Wind turbines are designed as special structures with low starting torque, and many starting torque minimization processes exist for permanent magnet synchronous generators (PMSGs). Plurality is applied to decrease the starting torque in radial flux permanent magnet disk generators. The most popular starting torque minimization method uses a magnet skew technique. When used at $20°$, this technique reduced starting torque by 4.72% (on load) under 500 rpm at 50 Hz for 120 min. By contrast, a PMSG with magnet skew conditions set at under $2°$ reduced electrical power by 3.86%. For high-speed PMSGs, magnet skew techniques affect the generation of heat in the coils (stator), with heat decrease at the middle of the coil, on its surface and between the coils at 2.90%, 3.10% and 2.40%, respectively. PMSGs were installed in vertical axis wind turbines (VAWTs), and heat generation in relation to wind speed and electrical power was assessed. Magnet skew techniques can be used in PMSGs to reduce staring torque, while skew techniques also reduce electrical power and heat generated at the stator.

**Keywords:** skewing magnet coil; starting torque; thermal in PMSG; vertical axis wind turbine

## 1. Introduction

Nowadays, electrical power generation using alternative energy sources, particularly wind current energy [1], has attracted increasing attention to reduce dependence on the decreasing supply of fossil fuels and ameliorate environmental pollution. Wind energy conversion systems (WECS) are classified into two types: geared and direct-drive. WECS have many advantages including high-efficiency relating to geared drives, higher drive durability and reliability, lower maintenance, lower vibration and reduction in noise [1,2]. To maximise these advantages, wind turbine producers have fabricated direct-drive WECS.

Direct-drive wind turbines must be economically viable compared to other power generation systems in terms of cost per unit torque force, torque density, weight of fluent materials, efficiency, overall volume, overall length, outer diameter, energy efficiency per charge, annual energy efficiency and overall cost. Energy cost as the bottom line was compared. Direct-drive electrical generator systems that focused on these criteria

indicated that permanent magnet (PM) machinery with many pole numbers was the most appropriate [3,4]. PM machinery can be classified into three types: transverse-flux, axial-flux, and radial-flux. Transverse-flux permanent magnet (TFPM) machinery operates as a high torque density force and leakage line flux that generates a poor power factor. Decreasing torque density of TFPM machinery should reduce the number of poles with low flux leakage. However, the main problems encountered by the rotating section of TFPM machinery are the difficult production methods and the large number of parts, resulting in a weak mechanical structure [5]. Currently, direct-drive (DD) electrical generators have become popular for wind turbines rated at 4 MW, and are also supplied for higher power wind turbines. The turbine blades transfer power directly to the DD electrical generator, thereby eliminating the gearbox. The DD technique has been widely adopted for economical energy supply by operating an electric generator at lower wind speeds using a reliable drive process [6]. Size reductions of wind turbine electrical generators are limited by the ability to radiate heat from the stator winding. Liquid cooling of the stator winding maximises electricity generation for smaller generator sizes compared to air-cooled generators of a similar size at optimal cost [7].

Cogging torque is important in the fabrication of PM machines, as the torque is required to overcome the opposing attractive force between magnets on the rotor and the iron teeth of the stator [8]. Cogging torque is sensitive at low-speed wind turbine applications. The PM wind generator eliminates cogging torque due to potential vibrations and noise to overcome starting turbine difficulty. Low-speed wind turbines focus on reducing cogging torque [9,10]. The starting speed of a wind turbine is an important factor for blade design to minimise cogging torque of the PM generator and friction on the shaft.

A wind turbine starting torque is generated by the blades under an exact wind speed. Blade movement must prevail over friction and cogging torque [11]. A large cogging torque is produced in a PM generator during start-up, and if the wind turbine fails to overcome this, it will not start [12]. The optimal cogging torque for a direct-drive PM generator was found to be between 1.5% and 2% of its minor torque [9]. Axial-flux permanent magnet (AFPM) machinery reduced the cogging torque by the appropriate values of magnetic pole arc on the pole pitch ratio [13]. The shape and amplitude occupied to the cogging torque signal depends on the magnetic pole arc and on the pole pitch ratio. Therefore, decreasing the magnet pole arc on the pole pitch ratio decreased magnet flux leakage and reduced the average torque [14].

Many cogging torque reduction methods have been studied, including adjusting the distance between magnets relative to each other, producing magnets with various pole arcs on to pole pitch ratios, distribution of slots and slot opening and connecting distributions of the rotors corresponding to each sector. The skewing technique of rotor magnets and stator slots is used to proficiently control cogging torque in AFPM machinery. Leakage inductance and copper losses increase when using skewing stator slots and the cogging torque of PM machinery decreases. The magnet skewing method using rounded magnets to form triangular shaped skewing magnets, dual skewed magnets and parallel-sided magnets can be adapted to decrease cogging torque in AFPM machines [15].

This research studied the relationship between the reduction of starting torque using the magnet-coil skewing technique at the rotor-stator that generated heat at the stator winding based on three positions as between the coils, on the surface of the coil and in the middle of the coil, as shown in Figure 1. The permanent magnet synchronous generator (PMSG) used the magnet-coil skewing was installed in a vertical axis wind turbine (VAWT) to collect data and investigate the relationship between heat in the PMSGs, wind speed and electrical power. Therefore, this investigation was research relationship of the magnet-coil skewing technique, reducing starting torque, temperature, and electrical power. The experimental laboratory was guided to installed real experiments (VAWT) for confirmed utilization in low speed wind turbine.

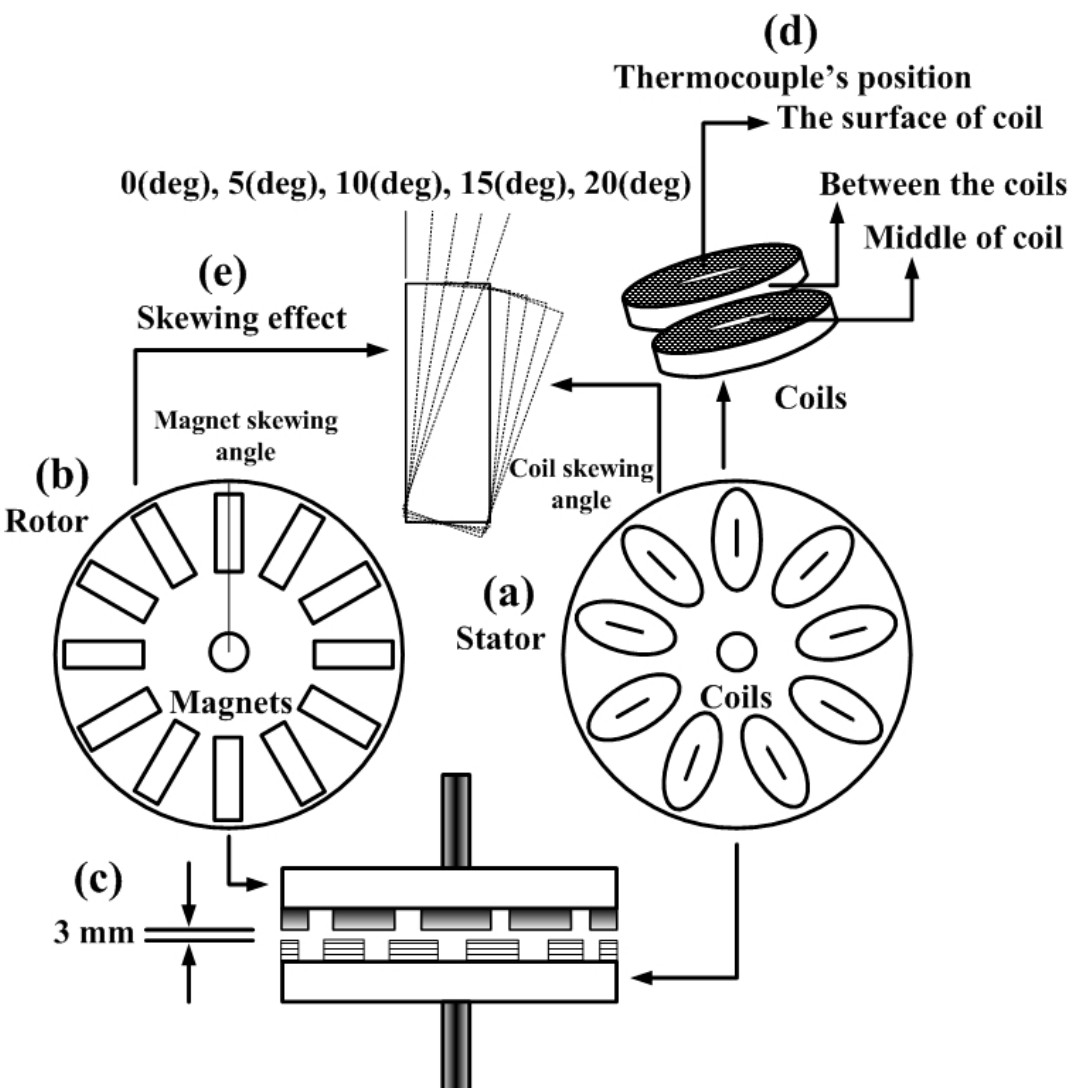

**Figure 1.** The PMSG disk: (**a**) Nine coils installed on Bakelite plates for the stator section; (**b**) Twelve magnets installed on Bakelite plates for the rotor section; (**c**) Permanent magnet synchronous generator (PMSG) installed with a 3 mm air gap; (**d**) K-type thermocouple for temperature conduction; (**e**) Magnet skewed between $0°$ and $20°$.

## 2. Experimental Study

This investigation used Bakelite plates for the 10 mm thick fabric of the rotor and stator. Specifications of the PMSG disk are shown in Table 1. Accessory of permanent magnets and disk rotor were attached on the rotor disc in order to adjust the skewing magnet and coil angle to decrease the starting torque, as shown in Figure 1. The air gap between the stator and rotor disc could be easily varied for the stator state, while the air coils were positioned on the stator disc with a connecting wire to produce a 3-phase PMSG [16]. The testing station operated a 3-phase 0.75 hp motor with a direct drive to the PMSG disk. The speed of the motor and its direction were powered by a 1-phase inverter. The measurement instruments included a torque meter (BCM Sensor Technologies, Model: 1811, Cap: 500 Nm, Accuracy (torque): 0.5% fs, Supply: $\pm15\ V_{dc}$, Max. speed: 7000 rpm, Load current: <10 mA). A K-type thermocouple was inserted between the air coils at the surface and middle of the air coil, PMSG generator was covered by transparent cast acrylic sheets for controlling variable between room and coils temperature. Measurements were conducted using a Petit Data logger GL100-WL, GS4-VT (Graphtec), 4 channel, sampling intervals for 5 min,

Temperature range at −200 to 400 °C, Accuracy type K ±0.05% and the power was recorded using a 3-phase Power & Harmonic Analyser (Lutron: DW-6095), frequency range at 40–60 Hz, watt hour range at 0.000 to 9.999 kWh with ± (2% + 0.008 kWh). The output of the PMSG was controlled by 450 W lamp load, as shown in Figure 2.

**Table 1.** Specifications of the PMSG disk.

| Item | Parameter | Value | Unit |
|------|-----------|-------|------|
| General data | Number of phases | 3 | − |
| | Winding resistance per phase | 45 | Ω |
| | Magnet | NdFeB | − |
| | Flux density | 0.251 | Tesla |
| | Air gap | 3 | mm |
| | Thickness of Bakelite plate | 10 | mm |
| Rotor | External diameter | 400 | mm |
| | Number of poles | 12 | − |
| | Magnet size | $100 \times 20 \times 5$ | mm |
| Stator | Thickness of Bakelite plate | 10 | mm |
| | External diameter | 400 | mm |
| | Copper coil size | $100 \times 70 \times 12$ | mm |
| | Number of coils | 9 | − |
| | Number of winding turns per coil | 900 | turns |

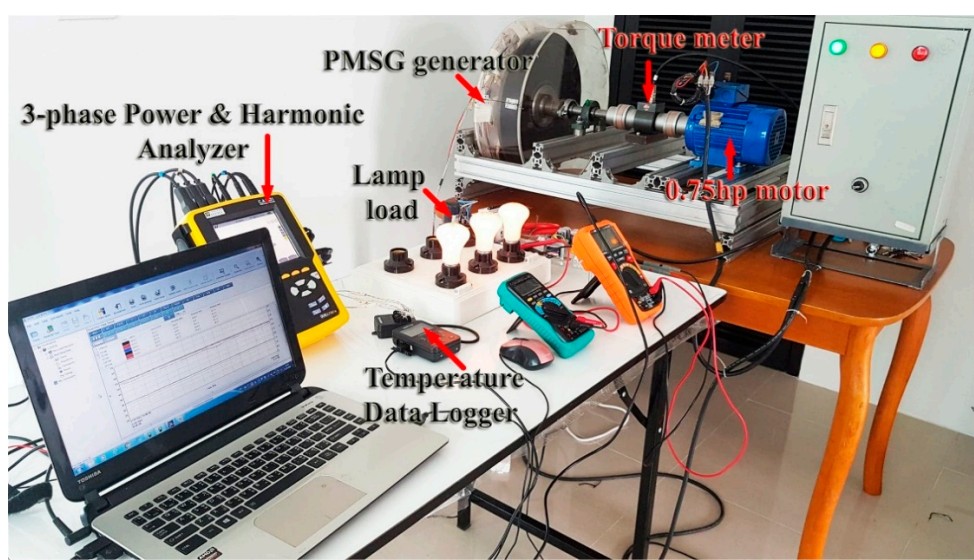

**Figure 2.** PMSG testing station.

The room temperature in the laboratory was 29 °C and the time duration was 120 min at a speed of 500 rpm. The atmospheric temperature of Rajjaprabha Dam was average 32 °C. Thermocouples were installed at three positions on a 10 kW vertical axis wind turbine (Rajjaprabha Dam, Surat Thani Province, Thailand) to measure average temperatures based on real operation of the wind turbine. The PMSG generator was equipped with 20° skewing magnets. Temperatures at three positions were recorded by a thermal data logger, K-type, GL100-WL (Graphtec), as shown in Figure 3. The weather station used a PROTRONICS, JEDTO: AW002, Outdoor temperature range: −40.0 °C to +65.0 °C, Wind speed: 0~100 mph. This instrument to measure the atmosphere temperature, wind speed and wind current direction.

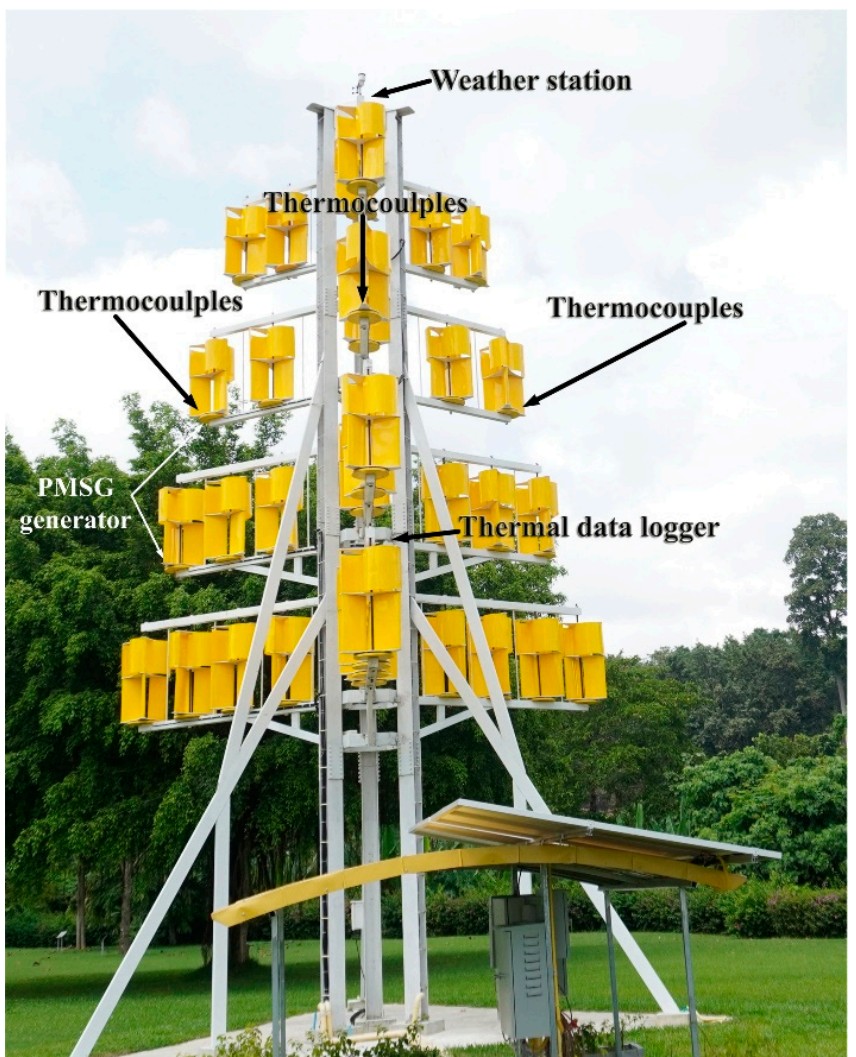

**Figure 3.** A 10 kW vertical axis wind turbine (Rajjaprabha Dam, Surat Thani Province, Thailand) with thermocouples installed on a PMSG generator to measure heat at the surface of the coils, between the coils and in the middle of the coils.

### 3. Results and Discussion

Figure 4a, shows the transient torque at starting time when the coils were skewed in the stator when operating at a rotation speed of 500 rpm. The coil skew was changed from 0, 5, 10, 15 and 20°, while the magnets in the rotor stood at 0°. The starting torque was increased up to 1.5–2.0 s, 1.3 Nm. Starting torque peak ranged from 4–7.5 s and stabilised after 7.5 s at 0.9 Nm. A 5° coil skew showed lower starting torque compared to the others, generating peak torque at 5.5–7.0 s with 1.25 Nm.

Figure 4b, shows the transient starting time when the magnets were skewed in the rotor when operating at a rotation speed of 500 rpm. The magnet skew was changed from 0, 5, 15 and 20°, while the coils in the stator were fixed at 0°. The starting torque increased up to 1.4–1.6 s, 1.3 Nm, with the peak starting torque at 4.5–7.5 s that stabilised after 7.5 s at 0.9 Nm. A 20° magnet skew gave a lower starting torque compared to the others, producing peak torque at 5.3–6.5 s with 1.23 Nm.

The starting torque is generated by the quantity of energy variation that corresponds to the rotation level of the rotor and can be indicated as follows:

$$T_{starting} = -\frac{\partial W(\alpha)}{\partial \alpha} \tag{1}$$

where $T_{starting}$, $W(\alpha)$ and $\alpha$ are the starting torque, flux in the air gap, and angle of rotor or stator, respectively. In a PMSG, the starting torque can be obtained by calculating the partial differential of the flux in the air gap areas related to the rotation angle of the rotor or stator [17].

$$T_{(\alpha)} = \frac{L_s \pi}{2\mu_0} \left( R_2^2 - R_1^2 \right) \sum_{n=0}^{\infty} G_{nN_L} B_{nN_L} \sin n N_L \alpha \qquad (2)$$

where $\mu_o$, $L_s$, $R_1$, $R_2$, $G_{nNL}$, $B_{nNL}$ and $N_L$ are the permeability of air, stack length, permanence magnet radius and stator radius corresponding to the air gap and flux density functions, respectively. The starting torque of each rotor or stator angle $T(\alpha)$ decreases if $G_{Nnl}$ or $B_{Nnl}$ are assumed to be zero, as shown in Equation (2) above.

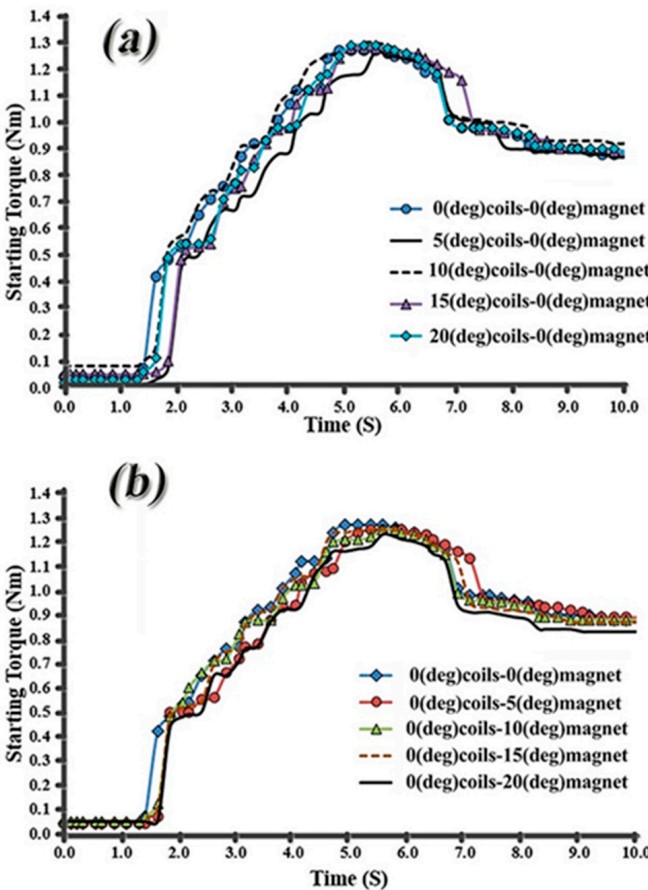

**Figure 4.** The effect of skew type on the starting torque in a PMSG operating at 500 rpm with 300 W electrical load: (**a**) Effect of coil skew angle; (**b**) Effect of magnet skew angle.

Figure 5 shows the comparison between 20° skewing magnet at the rotor and 5° skewing coils at the stator that change the areas of AB, BC and CD overlapped in the generator. $G_{nNL}$ can be represented in Equation (3) as follows:

$$G_{nN_L} = \frac{1}{n\pi} \frac{N_s}{N_P} \left( \sin n \; N_L a_{skewing} + \sin n \; N_L b_{skewing} + \sin n \; N_L c_{skewing} \right) \qquad (3)$$

Based on the reduction of $G_{NL}$ level among values of $Gn_{NL}$, where $n$ is 1, this skewing technique can be eliminated as a notable component of the starting torque. $N_P$ is the number of magnets and $N_S$ is the number of coils. The skewing magnets and coil widths $AB_{skewing}$, $BC_{skewing}$ and $CD_{skewing}$ are shown in Equation (4) below, and a notable component of the starting torque can be reduced [17].

$$\sin n \, N_L a_{skewing} + \sin n \, N_L b_{skewing} + \sin n \, N_L c_{kewing} = 0 \tag{4}$$

**Figure 5.** Comparison of overlap area for 0° to 20° skewing magnet and 0° to 5° skewing coil.

Skewing magnets and coils with widths of $AB_{skewing}$, $BC_{skewing}$ and $CD_{skewing}$ gave the maximum $G_{NL}$ value. The skewing position of the rotor (or stator) changed the areas for negative and positive maximum flux linkage of coils that affected $G_{NL}$ [18]. The 20° skewing magnets and 5° skewing coils reduced $G_{NL}$ values, as shown in Figure 4a,b. The $N_C$ value indicates the starting torque per rotation for a rotor fabricated in the stator, as shown in Equation (5) [19].

$$N_c = \frac{N_P \, N_s}{HCF \, (N_P, N_S)} \tag{5}$$

where *HCF* $(N_P, N_S)$ is the peakest common factor of the number of coils and magnets. Thus, the mechanical angle ($\alpha_c$) at which starting torque generates once can be written as Equation (6) as follows:

$$\alpha_c = \frac{360°}{N_c} \deg(m) \tag{6}$$

Hence, composing this as an electrical angle ($\alpha_{ce}$) gives Equation (7).

$$\alpha_{ce} = \frac{P}{2} \alpha_c \deg(e) \tag{7}$$

Then, the stator contribution transition angle to move the starting torque phase by 180 deg (*e*), generated in the coils, can be shown as Equation (8) below.

$$\alpha_{skewing} = \frac{1}{2}\alpha_{ce} = \frac{1}{2}\frac{p}{2}\alpha_c = \frac{1}{2}\frac{p}{2}\frac{360°}{N_c} = 90°\frac{HCF(N_PN_s)}{N_s}\deg(e) \tag{8}$$

$N_P$ and $N_S$ of the disk-PMSG relate to 12 magnets and 9 coils, respectively. Therefore, the optimal value of magnet coil skewing ranged from 0° to a maximum of 20°, as shown in Figure 5.

Figure 6 compares the magnet and coil skewing angle conditions that affected temperature, starting torque and average electrical power (W) load. Based on the temperature and magnet skewing angle condition of 0–20°:

- The temperature in the middle coils ranged from 47–48 °C.
- The temperature at the surface of the coils ranged from 43–45 °C.
- The temperature between the coils ranged from 40–42 °C.
- All three temperature ranges gradually decreased when skewing the magnets. For a coil skewing angle of 0–20°:
- The temperature in the middle coils ranged from 47–48 °C, with a higher temperature trend of 48 °C at a 10° skewing angle.
- The temperature at the surface of the coils ranged from 43–45 °C, with a higher temperature trend of 45 °C at a 5° skewing angle.
- The temperature between the coils ranged from 41–43 °C, with a non-linear trend at 43 °C at a 10° degree skewing angle.

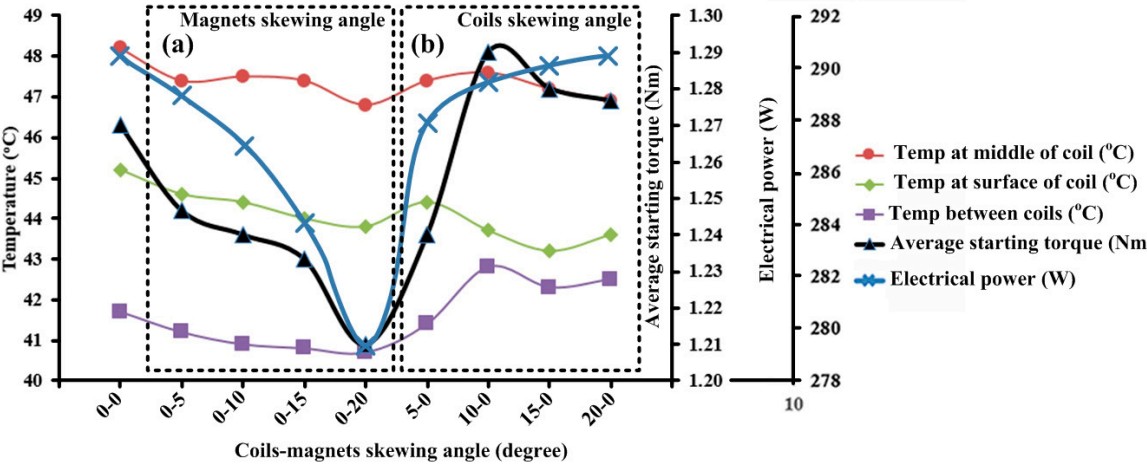

**Figure 6.** Comparison of temperature at the magnet coil average starting torque and average power for generator operation at 500 rpm: (**a**) Magnets skewing angle; (**b**) Magnets skewing angle.

All trends were non-linear in behaviour. The main point in thermal modeling is the analysis of convective heat transfer in a disc-type electrical generator. The heat flux of the isothermal surface $i$, illustrated in Figure 1d, is defined by Equation (9) [20]:

$$q_i = h_i\left(T_{surf,i} - T_{ref,i}\right) \tag{9}$$

where $q_i$ is the heat flux, $h_i$ is the average value of the surface area convective heat transfer coefficient $i$, $T_{surf,i}$ is the temperature of the surface $i$, and $T_{ref,i}$ is the average bulk fluid temperature of a closed volume $V_i$. The stator is ironless and all the losses are in the copper coils. Both load and no-load losses need to be investigated. No-load losses are generated by the activation of a rotating magnetic field inducing eddy and circulating currents in all coil sectors. Circulating currents act in the parallel-connected coils due to the induction of voltage distinctions between the coils. These coils behave like additional Joule heating in the winding, even when the generator is operated at no-load. Finally, the phase currents generated $I^2R$ loss in the winding [21]. A small proportion of losses

occurred in the rotor (magnets) compared to the stator, although this only happened when the generator was loaded.

Based on the speed rate, the rotor losses accounted for 16% of the entire electrical losses. The electrical field from the winding fabricated asynchronous time-varying fluxes into the rotor that caused eddy current losses in the magnet. However, the hysteresis losses were negligible at low electric frequencies [22]. The eddy current in the winding can be estimated using the well-known Carter formula [21], as shown in Equation (10) below. This is highly accurate at low electric frequencies (i.e., $f \leq 50$ Hz) [23]. Copper and rotor losses were computed as shown in Equations (11) and (12) below.

$$P_{eddy\_coils} = \frac{\left(\pi^3 B_g^2 f^2 L_m N_t\right)}{4\rho C_u (1 + \alpha C_u \Delta T_{coil})} \tag{10}$$

where $P_{eddy\_coil}$ is eddy current loss, $B_g$ is the peak level of air gap flux density caused by the permanent magnets, $L_m$ is the mean turn length occupied by the magnet, $N_t$ is the number of turns per coil, $\rho C_u$ is the density of copper coils, and $\alpha C_u$ is a copper thermal resistivity coefficient.

$$P_{Cu} = \left(3I_{rms}^2 R_{Cu}^{amb}\right) / (1 + \alpha_{Cu}\Delta T_{coil}) \tag{11}$$

$$P_{rotor} = \left(3I_{rms}^2 R_{core}\right) / (1 + \alpha_{core}\Delta T_{core}) \tag{12}$$

where $P_{CU}$ is copper losses in coils (W), $P_{rotor}$ is magnet losses in rotor (W), $I_{rms}$ is phase rms current (A), $R_{cu}{}^{amb}$ is copper phase resistance at ambient temperature ($\Omega$), $R_{core}$ is core resistance ($\Omega$), $\alpha_{Cu}$ is the copper thermal coefficient, $\alpha_{core}$ is the core thermal coefficient, $\Delta T_{coil}$ is the temperature rise in coils over ambient temperature (°C), and $\Delta T_{core}$ is the temperature rise in the core over ambient temperature (°C). The core resistance $R_{core}$ combines the magnet and ironless coils, as shown in Figure 1. Therefore, the thermal resistivity coefficients of the iron and neodymium magnets were around 300 and 4000 times lower than the copper wire that generated the thermal conducting range of 0–120 °C [21], respectively. Using this system, variations in resistivity in the iron and magnets can be neglected, with rotor losses simply calculated as $P_{rotor} \approx 3I^2{}_{rms}R_{core}$ [21]. Therefore, skewing the magnet and coils affected the temperature of the coils in the stator of the generator, and this changed the heat flux ($h_i$), eddy current losses ($P_{eddy\_coils}$), copper losses in coils ($P_{CU}$), and magnet losses in rotor ($P_{rotor}$) in relation to Equations (9)–(12).

Figure 6a,b shows that temperature in the middle of the coils was highest and caused by rapid heat flux collection from the copper coils, corresponding to Equation (9). The skewing magnets range of change of 0–20° decreased the temperature at all three points on the coil, as the neodymium magnet slightly changed in resistivity when operated by a generator. Temperatures increased between the skewing coils at 10–20° as the surface between the coils was closed and fitted. The cumulative heat flux around the coils is shown in Equation (9). Moreover, the skewing coils were easily affected by the density of copper coils ($\rho C_u$) and the copper thermal resistivity coefficient ($\alpha C_u$) that accumulated eddy currents in the coils, as shown in Equation (10), generated higher temperatures.

Figure 6a shows the average starting torque compared with a magnet skewing angle of 0–20°. The trend of starting torque was located in the range 1.21–1.27 Nm. Magnet skewing 0–20° slightly decreased the starting torque. This graph had a similar trend to the temperature graph. The average starting torque compared with the angle was also in the range of 0–20° skewing coils and the trend of starting torque was located in the range of 1.24–1.29 Nm, as shown in Figure 6b. The 5° skewing coils were reduced by 4.33% starting torque when compared with 0° skewing coils.

The average starting torque can be explained by Equations (1) and (2), depending on the flux in the air gap $W(\alpha)$ and the angle of rotor or stator ($\alpha$). Skewing magnets and coils were variably affected by $W(\alpha)$ and $\alpha$ [17]. Therefore, the average starting torque was linearly reduced, corresponding to the 0–20° skewing magnet, as shown in Figure 6a. The

skewing magnet changed the stability of the flux in the air gap. The indicated average starting torque fluctuated in the range 279–290 W when changed to 0–20° skewing coils, as shown in Figure 6b. However, 5° skewing coils gave the lowest torque as this position reduced the flux in the air gap to generate the average starting torque.

Figure 6a, shows the trend of average electrical power when the skewing magnet varied by 0–20°. This reduced linearly from a maximum of 290 W to a minimum of 279 W. Figure 6b shows the trend of average electrical power when skewing coils varied by 0–20°. This increased to a maximum of 290 W. Magnet-coil skewing was affected by field phase shift between points along the axis of the generator that reduced the induced electrical power in the PMSG. The electrical power depended on $P_{eddy\_coils}$, $P_{cu}$ and $P_{rotor}$ when magnet-coil skewing changed shape. This affected $I_{rms}$ inside the stator coils, as shown in Equations (11) and (12). Skewing coils at 5–20° increased the electrical power caused by the fitting density of copper coils ($\rho C_u$). The skewing magnet decreased the degree of skewing angle but did not affect the density of the copper coils [23] that only changed the flux through to the coils.

Figure 7 shows the relationship between rotating speed and electrical power by comparing the optimal conditions of skewing by 0° coils and 0° magnets, and 20° magnets and 5° coils. The skewing magnets and coils were impacted by high electrical power, with speeds of 400, 450 and 500 rpm relating to 200, 250 and 300 W, respectively. Skewing 20° magnets at 400, 450 and 500 rpm reduced electrical power by 3.86%, while skewing 5° coils at 400, 450 and 500 rpm reduced electrical power by 0.83%. Conversely, a low speed between 50 and 350 rpm indicated almost the same values.

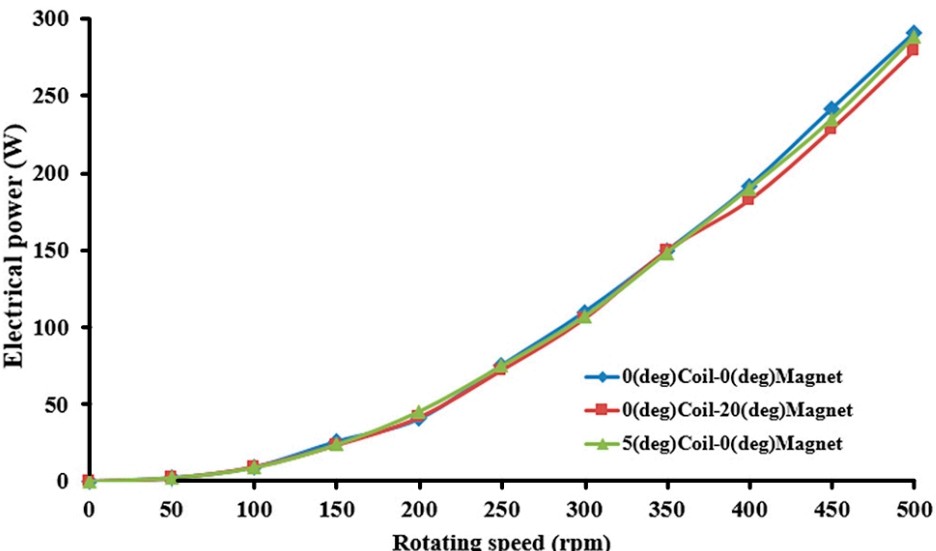

**Figure 7.** Relationship between rotating speed and electrical power comparing the optimal skewing conditions of 0° coils–0° magnets, 0° coils–20° magnets and 5° coils–0° magnets.

Torque level and mechanical speed were conducted freely at the shaft of the electric generator in steady-state circumstances. At a low-speed condition, mechanical losses were linear in proportion to the rotational speed, while circulating current and eddy losses were exactly proportional to the square of rotation speed. Mechanical and electrical losses are defined in Equations (13) and (14).

The constant values of $k_m$ and $k_{eddy}$ were fitted to experimental data in Figure 6, while the circulating current coefficient $k_{circ}$ was conducted from the loss segregation [21].

$$P_{mech-loss} = K_m \omega_m \tag{13}$$

$$P_\omega = \left( k_{eddy} + k_{circ} \right) \omega_m^2 \tag{14}$$

where $P_{mech\text{-}loss}$ is the mechanical loss (W), $P_\omega$ is loss due to eddy and circulating currents in the coils (W), $K_m$ is a constant for mechanical losses (Nm), $\omega_m$ is angular velocity (rps) then 1 rps = 60.000000024 rpm, $k_{eddy}$ is a constant for eddy current losses (Nm (rad/s)$^{-1}$), and $k_{circ}$ is a constant for circulating current losses (Nm (rad/s)$^{-1}$). Segregation between eddy and circulating current losses is plausible by representing the no-load test twice, with parallel connection and disconnected coils. Moreover, mechanical loss increases had no impact on the electrical heat generated [21]. Therefore, the skewing coils reduced electrical power at a higher rate than the skewing magnets due to eddy and circulating currents in the coils at high speeds of between 350–500 rpm.

Figure 8 shows a comparison between electrical power from a 10 kW VAWT and the temperature in a PMSG generator under variable wind speeds at Rajjaprabha Dam, Surat Thani Province, Thailand, during December 2019. Figure 8a shows the average electrical power data as real measurements from the PMSG of the VAWT, while Figure 8b shows the average electrical power maximum (2.4 kWh) and minimum (1 kWh), and the comparison between the temperature in the coils at three positions at average wind speed. The wind speed was a maximum of 4 m/s and a minimum of 2.7 m/s [24], which elevated the temperature at the middle, surface and between coil positions. Particularly, 4th and 16th December were the lower temperature of month leading to the low wind speed and low power generation. The atmospheric temperature is obviously affected data collection such as sun shine and raining. Therefore, the temperatures was maximum 36 °C and minimum 28 °C. The wind speed turned the blades that drove the PMSG generator to produce heat flux around the coils [20]. Therefore, the wind speed impacted the temperature around the coils and skewing magnets, and reduced the temperature in the coils.

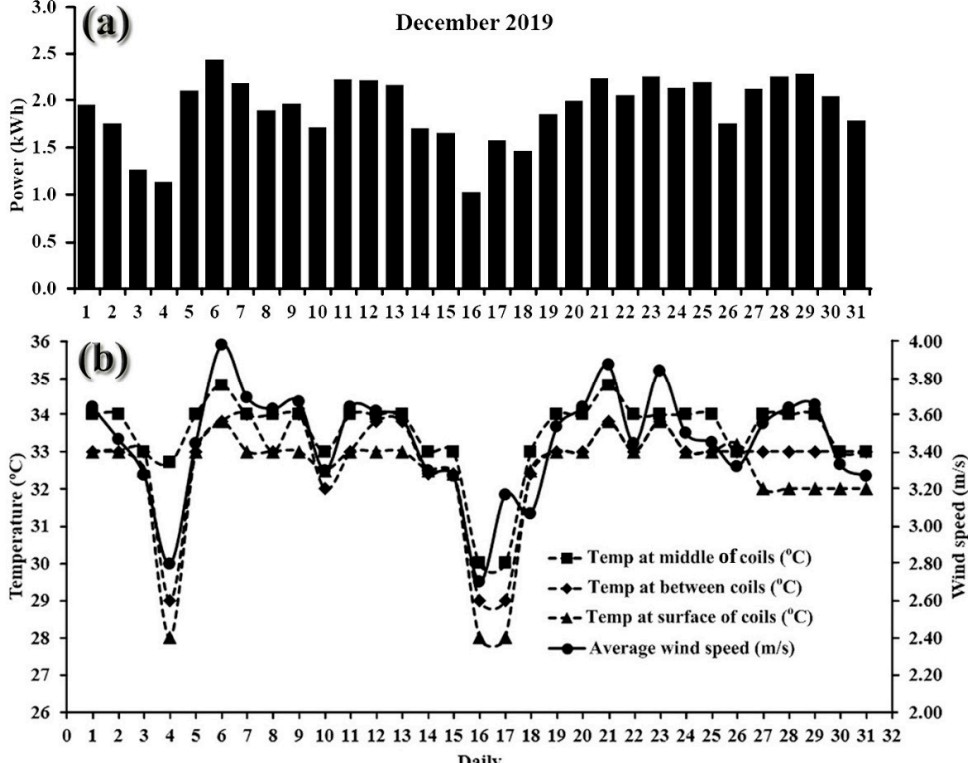

**Figure 8.** PMSGs installed in VAWTs at Rajjaprabha Dam, Surat Thani Province, Thailand. Data collected in December 2019 compared temperature, wind speed and electrical power: (**a**) average daily electrical power generation and (**b**) the relationship between wind speed and coil temperature.

## 4. Conclusions

PMSG generators are popularly fabricated for small wind turbines due to their low complexity, low cost and easy maintenance. The main generator components are a magnetic rotor and a coiled stator comprising a very basic electrical generator. Magnets provide a constant flux through to the coils that generate strong torque at the starting operation and temperature in coils of the PMSG generator. This study investigated and identified the reduction of starting torque and temperature by the skewing technique that can be used for both magnets at the rotor and coils at the stator. Heat was generated at the coils that affected the skewing magnets and coil behaviour.

Results indicated that 20° skewing magnets linearly reduced the heat at the coils, with maximum reduction of 2.90% at the middle of the coils, 3.10% at the surface of the coils and 2.4% between the coils. The starting torque reduced by a maximum of 4.72% at 20° skewing magnets, while electrical power reduced by a maximum of 3.86%. While the 5° skewing coils non-linearly reduced heat at all three coil positions, heat was reduced by a maximum of 0.72% between the coils, starting torque was reduced by a maximum of 4.33%, and electrical power was reduced by a maximum of 0.83%. All positions were compared to magnet coil 0° skewing. According to the studied in the real experiment installed in the 10 kW vertical axis wind turbine that used 20° skewing magnet caused the skewing magnet was easily installed in wind turbine. This investigation indicated that VAWT can be activated at cut-in 2.2 m/s, maximum power of 2.4 kW and minimum power of 1kW. The temperature in coils are fluctuated due to in the wind turbine was open systems which affected atmospheric temperature such as sun shine and raining. Hence, the temperatures was maximum 36 °C and minimum 28 °C.

The skewing coils also affected the starting torque, heat and electrical power. They were more affected by heat flux ($h_i$), eddy current losses ($P_{eddy\_coils}$), copper losses in coils ($P_{CU}$) and magnet losses in the rotor ($P_{rotor}$) than the skewing magnets that caused changes in the shape of the coils. Skewing magnets and coils reduced electrical power at high speeds of 400-500 rpm. The skewing technique can easily be used to reduce the starting torque and heat but the electrical power also decreases. Therefore, the PMSG generator can be applied for low-speed vertical axis wind turbines (VAWTs).

**Author Contributions:** Conceptualization, M.T., M.L. and T.T.; methodology, M.T. and M.L.; software, M.T., M.L. and C.K.; resource, M.T., M.L. and K.T.; validation, M.T. and M.L.; formal analysis, M.T., M.L. and T.T.; writing—original draft preparation, M.T. and M.L.; writing—review and editing, M.T., M.L. and T.T. All authors have read and agreed to the published version of the manuscript.

**Funding:** This investigation was financially supported by the Electricity Generating Authority of Thailand: EGAT (61-F405000-11-IO. SS03F3008347).

**Data Availability Statement:** The data presented in this study are available on request from the Corresponding author.

**Acknowledgments:** We respectively thank our colleagues from the Wind Energy and Energy Storage Systems Centre (WEESYC), the Interdisciplinary Graduate School of Energy Systems, the Centre of Excellence in Materials Engineering (CEME), the Energy Technology Research Centre (ETRC) and Faculty of Environmental Management, Faculty of Engineering, Prince of Songkla University, Thailand. Sincere thanks are also due to Wiriya Thongruang for his support and the use of his laboratory.

**Conflicts of Interest:** The authors declare no conflict of interest.

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
