# Peer review of "Relationship between Starting Torque and Thermal Behaviour for a Permanent Magnet Synchronous Generator (PMSG) Applied with Vertical Axis Wind Turbine (VAWT)"

_sustainability, doi:10.3390/su13169151_

Round 1
Reviewer 1 Report
Dear authors,
The strengths of your study are:
- the subject of the research;
- the results in laboratory conditions;
- presentation of a 10 kW VAWT functioning in real conditions.
In my opinion the manuscript needs improvements in every section.
- in Introduction – authors should focus on the "state of the art" of the subject, and present what aimed them to conduct in this research.
- in the “Experiment setup” section it is not clear what authors are testing in laboratory conditions and what in real conditions for the 10 kW VAWT.
- the discussion of findings needs improvement - there is lack of discussion regarding the obtained results for the 10 kW VAWT.
- the Conclusions section should be revised. All the conclusions presented are based on the results obtained in the laboratory, but there is no mention about the 10 kW VAWT.
- the References could be updated – most of the cited references have more than 10-15 years
- the English language needs revision.
Point by point observations / questions / recommendations are addressed as follows:
- Did authors run tests, in the laboratory, at different room air temperature to check the variations of the coil’s heat production?
- There are missing some technical specifications and accuracies of the instruments used (Petit Data logger 111 GL100-WL, Lutron: DW-6095, JEDTO: AW002)
- When detailing equation terms, authors should include the measuring units (where is suited) like they did at eq. 11 and 12.
- The sentence “NP is the number of magnets and NS is the number of coils.” Should be moved after equation 3 or start a new paragraph.
- Figure 6 should be completed. There are missing explanations for the right Y axis.
- The sentence “This reduced linearly from a minimum of 279 W to a maximum of 290 W.” should be reworded ….. maybe “This reduced linearly from to a maximum of 290 W to a minimum of 279 W.”
- Authors haven’t mentioned anything about the outside conditions (like temperature) when they run the real condition experiments.
- Figure 8 should be more detailed explained. For example, in 4 and 16 of December the outside conditions changed (temperature drops) leading to a smaller power generation by the VAWT.
- Based on what assumptions authors concluded that the wind speed elevated the temperature at the middle, surface and between coils position? From the figure 8?
- It is not clear if the results of the study presented in Conclusions are based on laboratory experiments? If so, what about real condition operations?
Author Response
Response to Reviewer 1 Comments
[Sustainability] Manuscript ID: sustainability-1332875 - Major Revisions
Point 1: Did authors run tests, in the laboratory, at different room air temperature to check the variations of the coil’s heat production?
Response 1: PMSG generator was covered by transparent cast acrylic sheets for controlling variable between room and coils temperature.
Point 2: There are missing some technical specifications and accuracies of the instruments used (Petit Data logger 111 GL100-WL, Lutron: DW-6095, JEDTO: AW002)
Response 2: The specifications were added in the new manuscript.
Point 3: When detailing equation terms, authors should include the measuring units (where is suited) like they did at eq. 11 and 12.
Response 3:
- aCu is the copper thermal coefficient (No unit)
- acore is the core thermal coefficient (No unit)
- DTcoil is the temperature rise in coils over ambient temperature (°C)
- DTcore is the temperature rise in the core over ambient temperature (°C).
Point 4: The sentence “NP is the number of magnets and NS is the number of coils.” Should be moved after equation 3 or start a new paragraph.
Response 4: Np and Ns have moved to the equation 3
Point 5: Figure 6 should be completed. There are missing explanations for the right Y axis.
Response 5: Labels of Y axis have added in Figure 6
Point 6: The sentence “This reduced linearly from a minimum of 279 W to a maximum of 290 W.” should be reworded ….. maybe “This reduced linearly from to a maximum of 290 W to a minimum of 279 W.”
Response 6: The sentence “This reduced linearly from a minimum of 279 W to a maximum of 290 W.” have changed to “This reduced linearly from to a maximum of 288 W to a minimum of 279 W.”
Point 7: Authors haven’t mentioned anything about the outside conditions (like temperature) when they run the real condition experiments.
Response 7: The atmospheric temperature of Rajjaprabha Dam average 32oC.
Point 8: Figure 8 should be more detailed explained. For example, in 4 and 16 of December the outside conditions changed (temperature drops) leading to a smaller power generation by the VAWT.
Response 8: Particularly, 4th and 16th December were the lower power of month leading to the low wind speed and low power generation. The atmospheric temperature is obviously affected data collection such as sun shine and raining.
Point 9: Based on what assumptions authors concluded that the wind speed elevated the temperature at the middle, surface and between coils position? From the figure 8?
Response 9: In the conclusion have added explanation of Figure 8 about temperature in coils
Point 10: It is not clear if the results of the study presented in Conclusions are based on laboratory experiments? If so, what about real condition operations?
Response 10: In the conclusion have added explanation of Figure 8 and explained the laboratory experiments was guided to selected the 20 degree used in real condition operations. And collected data followed real weather on December 2019.
Reviewer 2 Report
The specific comments are as follows:
(1) The title of the paper is too long. Make it concise.
(2) Clearly state the research gaps and novelty of the work.
(3) Section 1 should be concluded with the contributions and benefits of the proposed work. The Fig. 1 without any description seems out of place here.
(4) The paper lacks the research method part. Where is the theoretical background of the work? Include this section with necessary mathematical formulation.
(5) The result section is long and unorganized. Divide it in sub-scenarios and then explain the findings.
Overall, the presentation of the paper needs a lot of improvement to be qualified as a journal paper. Pls see the commonly used pattern for a journal paper from published articles and improve accordingly.
Author Response
Response to Reviewer 2 Comments
[Sustainability] Manuscript ID: sustainability-1332875 - Major Revisions
Point 1: The title of the paper is too long. Make it concise.
Response 1: The title have changed to shorter “Relationship between starting torque and thermal behaviour in Permanent Magnet Synchronous Generator (PMSG) applied Vertical Axis Wind Turbine (VAWT)”
Point 2: Clearly state the research gaps and novelty of the work.
Response 2: This article was combined the investigation between laboratory and real experiment. The experiment in laboratory can be guided for selected 20 degree skewing magnet using the real experiment in VAWT. This article was studied skewing magnet – coil technique, starting torque, temperature in coils, electrical power and wind speed that easily understood the relationship and confirmed utilization in smaller VAWT wind turbine.
Point 3: Section 1 should be concluded with the contributions and benefits of the proposed work. The Fig. 1 without any description seems out of place here.
Response 3: Figure 1 was explained position of thermocouple at between coils, surface of coil, and middle of coil. Figure 1 was also explained the how to use magnet-coil skewing technique.
Point 4: The paper lacks the research method part. Where is the theoretical background of the work? Include this section with necessary mathematical formulation.
Response 4:
- This investigation used many section for experiments (Research method part) that added explanation in “Introduction” and “Experimental study”
- The mathematic formulation is equation (1-2) that is the background of starting torque, temperature in coils is related in equation (11-12)
Point 5: The result section is long and unorganized. Divide it in sub-scenarios and then explain the findings..
Response 5: Results and conclusion were briefed followed reviewer’s comments
Point 6: Overall, the presentation of the paper needs a lot of improvement to be qualified as a journal paper. Pls see the commonly used pattern for a journal paper from published articles
Response 6: All the presentations of the paper was edited followed the comments of reviewers. And proofed by native speaker
Round 2
Reviewer 1 Report
Dear authors,
Your manuscript is visible improved.
In authors responses, you stated that you have rephrased the sentence “This reduced linearly from to a maximum of 288 W to a minimum of 279 W“, but in the revised manuscript it’s written “This reduced linearly from a maximum of 279 W to a minimum of 290 W” – correct the sentence
Reviewer 2 Report
My comments are addressed.